# Regulation of tumor angiogenesis and mesenchymal–endothelial transition by p38α through TGF-β and JNK signaling

Raquel Batlle [1], Eva Andrés[1], Lorena Gonzalez[1], Elisabet Llonch[1], Ana Igea[1], Núria Gutierrez-Prat[1], Antoni Berenguer-Llergo [1] & Angel R. Nebreda[1,2]

The formation of new blood vessels is essential for normal development, tissue repair and tumor growth. Here we show that inhibition of the kinase p38α enhances angiogenesis in human and mouse colon tumors. Mesenchymal cells can contribute to tumor angiogenesis by regulating proliferation and migration of endothelial cells. We show that p38α negatively regulates an angiogenic program in mesenchymal stem/stromal cells (MSCs), multipotent progenitors found in perivascular locations. This program includes the acquisition of an endothelial phenotype by MSCs mediated by both TGF-β and JNK, and negatively regulated by p38α. Abrogation of p38α in mesenchymal cells increases tumorigenesis, which correlates with enhanced angiogenesis. Using genetic models, we show that p38α regulates the acquisition of an endothelial-like phenotype by mesenchymal cells in colon tumors and damage tissue. Taken together, our results indicate that p38α in mesenchymal cells restrains a TGF-β-induced angiogenesis program including their ability to transdifferentiate into endothelial cells.

[1] Institute for Research in Biomedicine (IRB Barcelona), The Barcelona Institute of Science and Technology, 08028 Barcelona, Spain. [2] ICREA, Pg. Lluís Companys 23, 08010 Barcelona, Spain. Correspondence and requests for materials should be addressed to A.R.N. (email: angel.nebreda@irbbarcelona.org)

New blood vessel formation is essential for normal development, and tissue repair. Yet, angiogenesis has also emerged as a key factor for inflammatory processes, tumor progression and metastasis. Once tumors grow beyond a few millimeters, the angiogenic program is turned on. Blood vessel formation requires proliferation, migration and structural modification of endothelial cells, which are stimulated in the microvascular niche. There is evidence suggesting the existence of different stem and progenitor cell types in the vascular wall with diverse differentiation potential in vitro[1], but much remains to be learnt about the precise identities of the endothelial cell precursors. Besides endothelial cells, the vascular wall contains pericytes, perivascular or mural cells that closely encircle endothelial cells in capillaries and microvessels, maintaining vessel integrity and normalizing new vasculature[1]. Perivascular cells have a mesenchymal origin, and some mural cells express markers of mesenchymal stem cells (MSCs), retaining osteogenic, myogenic, adipogenic, and chondrogenic potential in culture[2–4]. There is also evidence that MSCs can differentiate into cancer-associated fibroblasts (CAFs) and myofibroblasts, which play important roles in the tumor microenvironment[3,5]. Several studies have reported the angiogenic potential of MSCs, through the release of factors that facilitate blood vessel formation[6,7], but whether these cells have the ability to acquire an endothelial phenotype is controversial[8,9].

The transforming growth factor-β (TGF-β) pathway exerts a tumor-suppressive function in cancer and is frequently inactivated by mutations in epithelial cancer cells, but elevated expression of *TGFB1* mRNA (encoding TGF-β) is associated with poor outcome in colorectal cancer patients[10]. This apparent controversy has been accounted for by an important role for TGF-β in the tumor microenvironment, which facilitates colorectal cancer progression and metastasis[10,11]. For example, TGF-β is a potent inducer of angiogenesis in vivo by modulating pro- and anti-angiogenic factors that affect both endothelial and mural cells[12].

Binding of TGF-β to its receptors induces phosphorylation of Smad proteins, the canonical mediators of TGF-β signaling, but can also activate other signaling pathways including the mitogen-activated protein kinases (MAPK) JNK and p38α[13]. The TGF-β-activated kinase 1 (TAK1) is essential for the TGF-β-induced activation of JNK and p38α and, interestingly, TAK1-deficient embryos presents vascular defects[14]. Signaling by p38α and JNK has been also linked to endothelial cell proliferation and apoptosis as well as to the production by endothelial cells of angiogenesis-regulation factors like VEGF[15–18]. However, the contribution of TGF-β-activated Smad and MAPK signaling to the conversion of MSCs to endothelial-like cells and whether this impinges on tumor angiogenesis has not been investigated.

Here we describe a new mechanism mediated by TGF-β/JNK signaling and negatively regulated by p38α that promotes angiogenesis and controls the fate of mesenchymal cells. We also provide evidence that mesenchymal cells may act as a source of endothelial cells during tissue repair and tumor angiogenesis.

## Results

**p38α negatively regulates blood vessel formation in tumors.** Angiogenesis is actively involved in tumor development. Studies in colon tumors from mouse models and patient derived xenografts (PDXs) have implicated p38α signaling in the regulation of tumor initiation and progression[19,20]. However, how p38α in cells of the tumor microenvironment contributes to tumor growth, and in particular to the angiogenic switch is poorly characterized.

During tumor-induced angiogenesis, endothelial cells and the surrounding pericytes that form the vasculature, develop multiple morphological and architectural abnormalities. To evaluate the role of p38α in tumor vasculature formation, we treated two PDX models of colon tumors with either the p38α inhibitor PH797804 or vehicle. Immunohistochemistry analysis using PDGFRB (CD140b) as a perivascular cell marker, and CD31 or CD105 as markers for mature or immature blood vessels, respectively, showed an enhanced number of blood vessels and perivascular cells in the colon tumors upon p38α inhibition (Fig. 1a).

To extend these observations, we used mice with the UBC-Cre-ERT2 transgene and floxed alleles of *Mapk14* encoding p38α. After 4-hydroxy tamoxifen (4-OHT) administration to induce p38α downregulation (p38αΔ^Ub), mice were treated with the carcinogen Azoxymethane (AOM) and three cycles of DSS[19] to induce colorectal tumors. Consistent with the PDX analysis, we found enhanced staining for CD31, CD105, CD34 (mature and immature blood vessel marker) and PDGFRB in colon tumors from p38αΔ^Ub mice (Fig. 1b), suggesting that p38α downregulation stimulated tumor vessel density and perivascular cell recruitment. Interestingly, double staining experiments revealed the co-expression of CD31 with PDGFRB or CD146 (M-CAM) in blood vessels of colon tumors from p38αΔ^Ub but not WT mice (Fig. 1c).

Taken together, these results suggest that p38α signaling negatively regulates new blood vessel formation during colon tumorigenesis, and that p38α-deficient perivascular cells co-express endothelial markers.

**p38α negatively regulates an angiogenic program in MSCs.** Pericytes that surround the endothelial cells play an important role in vasculature architecture, and pericytes with MSC traits have been identified in several human organs[2,3]. MSCs have been isolated from virtually every postnatal and fetal tissue, and have been shown to differentiate into different cell types in vitro[21].

Since p38α signaling can modulate cell differentiation programs[22], we investigated the role of this pathway in MSC fate. We generated immortalized MSCs from bone marrow of UBC-Cre-ERT2;*Mapk14*^lox/lox mice. Fluorescence-activated cell-sorting (FACS) confirmed that the isolated MSCs were negative for the endothelial markers CD31 and Ve-Cadherin (CD144) and the hematopoietic marker CD45 but were positive for several MSC markers including PDGFRB, CD29, CD105 and CD73, and also showed low expression of the perivascular marker CD146 (Supplementary Fig. 1a). Moreover, WT MSCs were able to differentiate into adipocytes and osteoblast (Supplementary Fig. 1b, c). Immortalized WT MSCs were treated with 4-OHT to delete p38α (p38αΔ), which was confirmed by immunoblotting (Supplementary Fig. 1d). WT and p38αΔ MSCs looked morphologically similar in subconfluent cultures. However, confluent p38αΔ but not WT MSCs showed a typical endothelial cobblestone phenotype (Supplementary Fig. 1e).

TGF-β is an important regulator of MSC activation and differentiation potential[5]. We found that TGF-β treatment induced p38α activation and enhanced phospho-Smad2 levels (Fig. 2a). To analyze the contribution of p38α to MSC functions, we performed transcriptomic analysis. Gene set enrichment analysis (GSEA) revealed a highly enriched signature of genes associated with angiogenesis and vasculature development in TGF-β-treated p38αΔ MSCs (Fig. 2b). Moreover, we confirmed that several angiogenesis-related genes were robustly upregulated in p38αΔ MSCs (Fig. 2c). Interestingly, the addition of TGF-β potentiated the changes in angiogenesis-related gene expression observed in p38αΔ MSCs suggesting that p38α signaling negatively regulates the TGF-β pathway in MSCs.

Mesenchymal cells can promote tumor growth by remodeling the extracellular matrix, recruiting endothelial cells and inducing

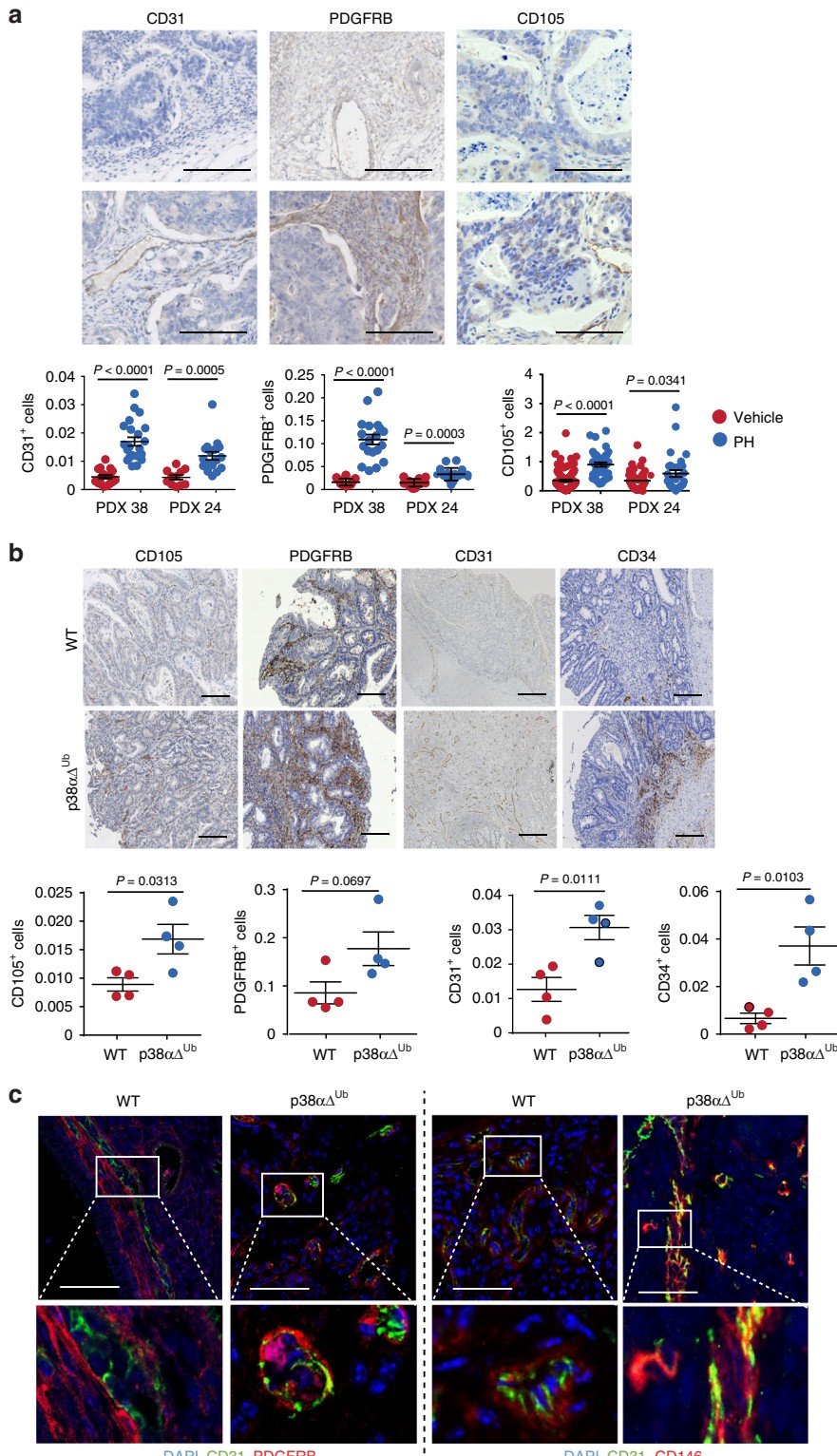

**Fig. 1** Pharmacological inhibition of p38α induces angiogenesis in human and mouse colon tumors. **a** Immunostaining analysis of two different human colon PDXs that were treated with the p38α inhibitor PH797804 (PH) or vehicle. The percentages of CD31+, CD105+, and PDGFRB+ cells among the total number of cells per tumor area were determined using ImageJ on pictures from colon tumors. $n \geq 4$ images for each condition. Bars, 100 μm. Data represent mean ± SEM ($n = 4$ tumors). **b** Immunostaining analysis of colon tumors induced by AOM/DSS in WT or p38αΔ^Ub mice. The percentages of CD31+, CD105+, CD34+, and PDGFRB+ cells among the total number of cells per tumor area were determined using ImageJ on pictures from three colon tumors for each condition. Bars, 100 μm. Data represent mean ± SEM ($n = 4$ mice). **c** Colon tumors induced by AOM/DSS in WT and p38αΔ^Ub mice were stained with DAPI (blue) and immunostained for the endothelial marker CD31 (green) and the perivascular markers PDGFRB or CD146 (red). Bars, 100 μm. The lower panels show higher magnifications of the indicated areas

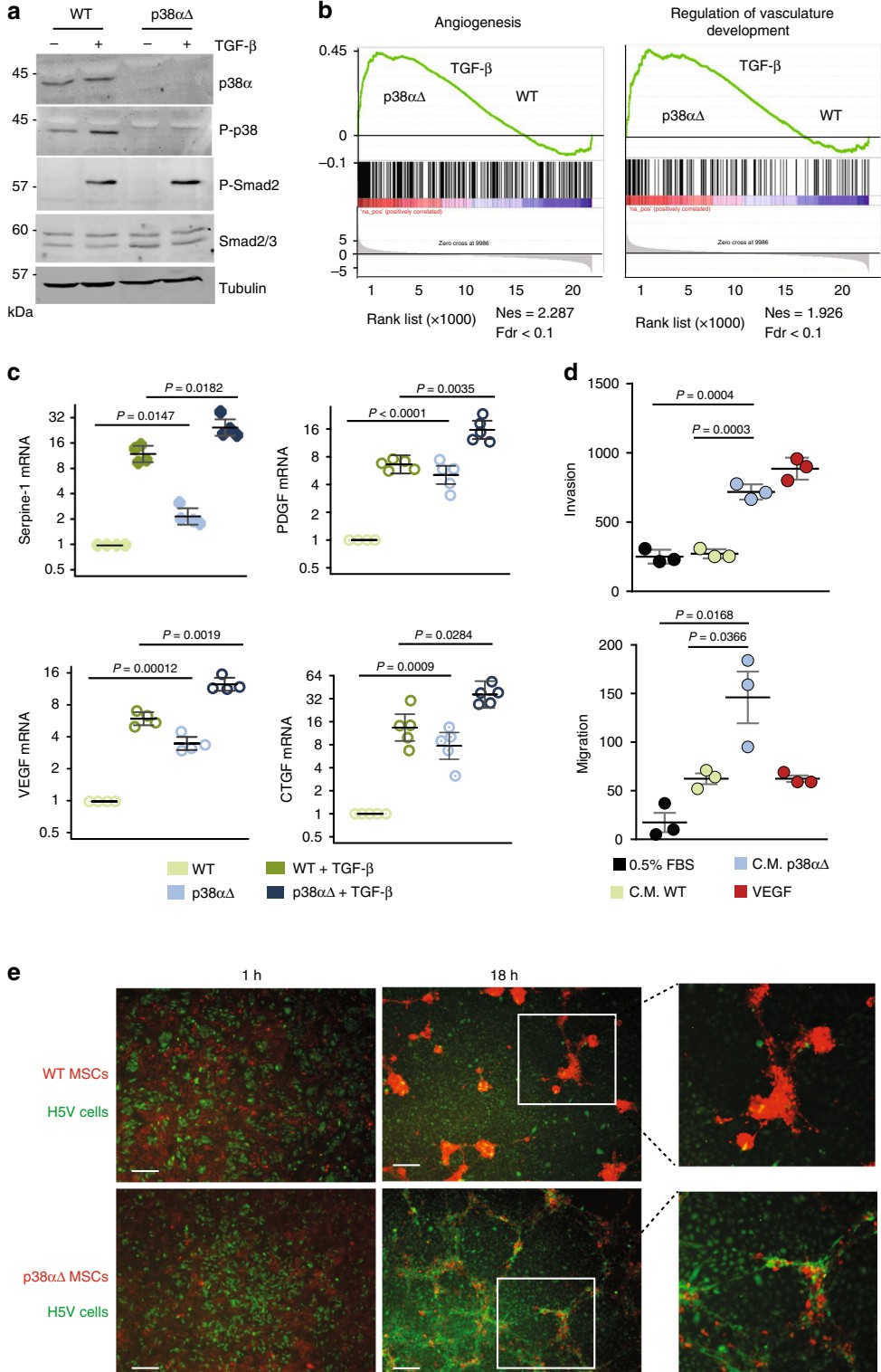

**Fig. 2** p38α negatively regulates TGF-β signaling and the expression of angiogenesis genes in MSCs. **a** MSCs were obtained from *Mapk14*[lox/lox];UBC-Cre-ERT2 mice, immortalized and then incubated with 4-OHT for 2 days to obtain p38αΔ MSCs. WT MSCs were treated with vehicle. Cells were starved (0.5% FBS) and treated with TGF-β for 20 h, and total protein lysates were analyzed by immunoblotting. **b** Gene set enrichment analyses (GSEA) of Angiogenesis (GO: 001525) and Regulation of vasculature development (GO: 1901342) comparing WT and p38αΔ MSCs treated or not with TGF-β. Nes normalized enrichment score, Fdr false discovery rate. **c** Relative levels of the indicated mRNAs in WT and p38αΔ MSCs treated with TGF-β. Values are graphically represented in log2 scale and show the fold change versus WT. Data represent mean ± SEM (n = 4) and CTGF (n = 5). **d** H5V endothelial cells were used for migration and invasion assays on Boyden chambers in the presence of 0.5% FBS, VEGF (10 ng/ml) or conditioned medium (C.M.) prepared from WT and p38αΔ MSCs after 48 h of culture. Quantifications were performed after 6 h for the migration assays or 16 h for the invasion assays by counting the number of crystal violet-stained cells in different fields. Data are means ± SEM (n = 3). **e** Co-cultures of H5V endothelial cells and MSCs pre-stained with cell tracker green and red, respectively, in DMEM 10% FBS for the indicated times. Bars, 100 μm. The right panels show higher magnifications of the indicated areas

vascularization during tumorigenesis[7]. We postulated that the upregulated angiogenic program in p38αΔ MSCs could facilitate migration and invasion of endothelial cells. Consistent with this idea, conditioned media collected from p38αΔ MSCs enhanced the migration and invasion of H5V endothelial cells (Fig. 2d). Furthermore, co-culture of H5V endothelial cells with p38αΔ MSCs but not with WT MSCs produced three-dimensional tubules that resembled small capillaries without the addition of exogenous extracellular matrix (Fig. 2e). These results indicate that p38α negatively regulates an angiogenic program in MSCs that releases pro-angiogenic factors and facilitates the migration and capillary formation by endothelial cells.

**p38α controls MSC differentiation into endothelial cells.** The differentiation of MSCs towards adipocyte, osteoblast and chondrocyte lineages is well established, and there is evidence indicating that MSCs can form capillary-like structures in vitro[8,9]. To investigate whether p38α controls the mesenchymal-to-endothelial transition, WT or p38αΔ MSCs were seeded on matrigel to facilitate capillary tube formation and subjected to serum starvation. Strikingly, we found that both primary and immortalized p38αΔ but not WT MSCs were able to form preliminary sprouting at 5 h, with well-formed capillary tube-like structures observed after 16 h (Fig. 3a and Supplementary Fig. 2a). Since other cell types, including fibroblasts and cancer cells, have been reported to form networks on matrigel[23], we confirmed that p38αΔ but not WT MSCs formed capillary tube-like structures in collagen matrices (Supplementary Fig. 2b). Importantly, endothelial cells are known to take up acetylated low-density lipoprotein (AcLDL)[24], and AcLDL added to the medium was detected 4 h later in the cytoplasm of p38αΔ MSCs but not in WT cells (Fig. 3b). Furthermore, we confirmed that primary and immortalized p38αΔ MSCs cultured on collagen were able to form a primary lumen that elongates along cell branches, as expected for new vessels (Fig. 3c, Supplementary Fig. 2c, and Supplementary Movies 1 and 2).

The capillary formation observed in p38αΔ MSCs correlated with enhanced expression of endothelial markers, as detected both by qRT-PCR and by immunohistochemistry (Fig. 3d and Supplementary Fig. 2d, e). Interestingly, TGF-β enhanced capillary tube formation in matrigel and collagen cultures (Fig. 3a and Supplementary Fig. 2b), and further increased the expression of endothelial markers by p38αΔ MSCs (Fig. 3d). On the other hand, genes associated to fibroblast fate were expressed at lower levels in p38αΔ MSCs (Fig. 3d). Treatment of WT MSCs with three different chemical inhibitors confirmed that p38α signaling negatively regulates the formation of capillary tube-like structures by MSCs (Fig. 3e and Supplementary Fig. 2f).

**TGF-β induces endothelial differentiation of p38αΔ MSCs.** Elevated TGF-β production in the tumor microenvironment has been associated with angiogenesis stimulation, contributing to tumor progression and metastasis[10,25]. There is evidence that MSCs differentiate into CAFs, and our data suggested that MSCs can also acquire an endothelial phenotype. To analyze the occurrence of MSC to endothelial cell differentiation in human colorectal tumors, we compared the TGF-β-induced gene expression program of MSCs with published gene expression signatures associated to endothelial cells (CD31[+]) or CAFs (FAP[+]) from human colorectal tumors[10]. To avoid biases, we only used genes that were overexpressed at least fivefold compared to the other cell populations represented in the tumors. The comparative analysis showed a strong enrichment of endothelial cell-associated genes among those overexpressed in TGF-β-

treated p38αΔ MSCs, but not in genes upregulated in WT MSCs (Fig. 4a). Overall expression of the CD31[+] signature was significantly higher in TGF-β-treated p38αΔ compared to WT MSCs Fig. 4b). In contrast, GSEA revealed that similar fractions of CAF-associated genes were overexpressed in p38αΔ and WT MSCs treated with TGF-β (Fig. 4a), and no significant differences were observed when the FAP[+] signature was evaluated (Fig. 4b). These results support that, among other transcriptomic alterations, TFG-β treated MSCs can acquire characteristics of bona-fide endothelial cells from human colorectal tumors in a process that is negatively regulated by p38α.

We investigated the mechanisms regulating the acquisition of endothelial characteristics by MSCs. We found that p38αΔ MSCs showed the upregulation of *Tgfb1, Tgfb2* and *Tgfb3* mRNAs, both under basal conditions and after TGF-β stimulation (Fig. 4c and Supplementary Fig. 3a), as well as increased production of TGF-β1 protein (Fig. 4c), suggesting that p38α signaling negatively regulates TGF-β production. GSEA also indicated that the TGF-β signaling signature was enriched in TGF-β-treated p38αΔ MSCs (Fig. 4d). Moreover, a Luciferase reporter under control of Smad-binding elements confirmed enhanced TGF-β signaling in p38αΔ MSCs both under basal condition and in response to TGF-β (Fig. 4e).

The regulation of angiogenesis by TGF-β involves two type I receptor/Smad pathways. ALK1 induces Smad1/5 phosphorylation and endothelial cell proliferation and migration, while ALK5 induces Smad2/3 phosphorylation and endothelial cell differentiation and blood vessel maturation[26,27]. We found that MSCs express both ALK1 and ALK5 but did not observe changes in their expression levels (Supplementary Fig. 3b). Moreover, TGF-β induced rapid Smad1/5 phosphorylation in WT MSCs, whereas this phosphorylation was reduced in TGF-β-treated p38αΔ MSCs (Supplementary Fig. 3c). In contrast, Smad2 phosphorylation was similar between WT and p38αΔ MSCs, but Smad3 phosphorylation was increased in p38αΔ MSCs after 10 h of TFG-β treatment (Fig. 4f). We also found increased accumulation of phospho-Smad3 but not phospho-Smad2 in the nucleus of TGF-β-treated p38αΔ MSCs (Fig. 4g and Supplementary Fig. 3d). Higher levels of phospho-Smad3 but not phospho-Smad2, were also detected under basal conditions in p38αΔ MSCs (Fig. 4h). Moreover, the mRNA for *Smad3*, but not *Smad2, Smad4* or *Smad7*, was upregulated in p38αΔ MSCs, either untreated or treated with TGF-β (Supplementary Fig. 3e).

TGF-β receptors phosphorylate the C-terminus of Smad transcription factors leading to their activation and TGF-β-induced gene expression changes. However, TGF-β can also induce the phosphorylation of additional serine and threonine residues within the central linker region of Smads, which elicits a variety of cell responses[28]. Interestingly, besides the canonical phosphorylation of the Smad3 C-terminus, we detected enhanced phosphorylation of Ser-208 in the Smad3 linker in p38αΔ MSCs (Fig. 4f).

Consistent with the idea that the TGF-β/ALK5 axis could mediate endothelial differentiation of p38αΔ MSCs, the chemical inhibitor LY2157299 abolished the formation of tubules in matrigel as well as the upregulation of endothelial markers in TGF-β-treated p38αΔ MSCs (Fig. 5a, b and Supplementary Fig. 3f). The implication of TGF-β in the conversion of MSCs to endothelial cells was confirmed using a neutralizing pan-TGF-β antibody (Fig. 5c and Supplementary Fig. 3g).

The results suggest that p38α prevents mesenchymal to endothelial differentiation by interfering with TGF-β/ALK5 signaling in MSCs, and identify TGF-β as a key regulator of the conversion of MSCs into endothelial cells.

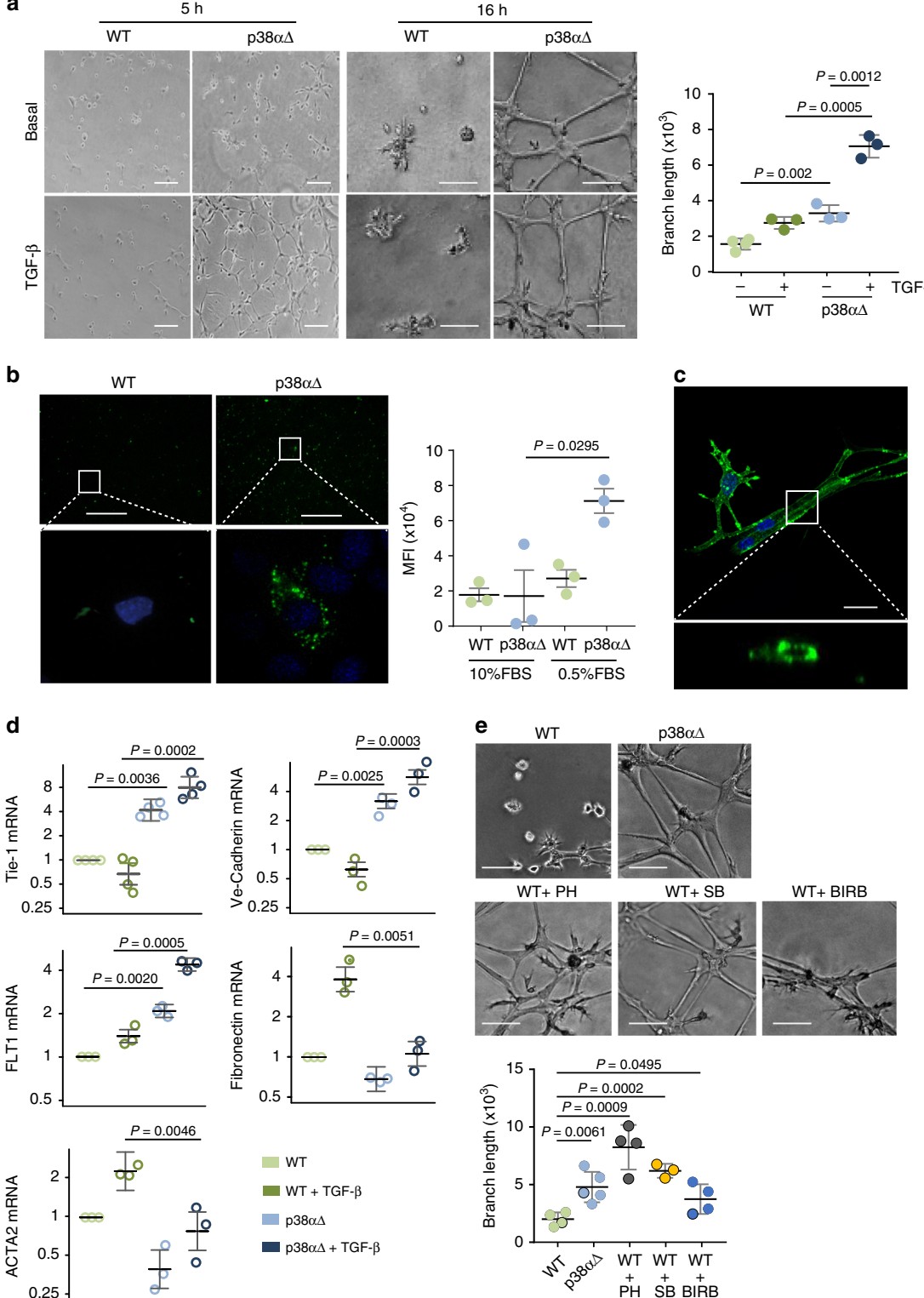

**p38α regulates TGF-β-induced angiogenesis though JNK.** Our results implicated TGF-β signaling in the mesenchymal-to-endothelial transition observed in p38αΔ MSCs, which correlated with increased Ser-208 phosphorylation in the Smad3 linker, indicating the activation of alternative non-Smad pathways. The Smad3 linker region contains several serine/threonine sites that can be potentially phosphorylated by MAPKs. It has been reported that p38α can negatively regulate JNK signaling[29], and we found increased levels of both active JNK and its effector

phospho-c-Jun in p38αΔ MSCs, both under basal conditions and in response to TGF-β (Supplementary Fig. 4a, b). TGF-β can induce activation of both JNK and p38α, which is thought to be mediated by TAK1. Moreover, TAK1 can be regulated by p38α through a negative feedback mechanism that limits p38α signaling and allows it synchronization with other TAK1-regulated pathways such as JNK signaling[30]. We found that ALK5 inhibition reduced the activation of both canonical and non-canonical TGF-β pathways (Fig. 6a). Moreover, TAK1 inhibition reduced

**Fig. 3** p38α-deficient MSCs adopt an endothelial phenotype ex vivo. **a** MSCs were seeded in matrigel with 0.5% FBS in the presence or absence of TGF-β (5 ng ml⁻¹) and tube formation was determined 5 h and 16 h later. The histogram shows the quantification of the branch length per field using an ImageJ macro at 5 h. Data are mean ± SEM ($n = 3$). Bars, 100 μm. **b** MSCs were starved and then incubated with AcLDL labeled with AlexaFluor488 in 0.2% FBS for 4 h. Green dots show AcLDL internalization in p38αΔ but not WT MSCs. The lower panels show higher magnifications of the indicated areas. The histogram shows the quantification of the mean fluorescence intensity (MFI) determined by scanning all fields with a ScanR inverted microscope. Bars, 500 μm. Data are mean ± SEM ($n = 3$). **c** p38αΔ MSCs were stained with Alexa Fluor 488 Phalloidin and cultured in a 3D collagen gel to allow formation of vascular structures with central lumen. The lower panel shows a higher magnification of the indicated area as the orthogonal projections along the vertical axis. Bar, 25 μm. **d** MSCs were cultured in 0.5% FBS for 7 days in the presence or absence of TGF-β, which was added to the medium every 2 days. Relative mRNA expression levels for the endothelial markers Tie-1, Ve-Cadherin and FLT1 (VEGFR1), and the fibroblast markers, Fibronectin and ACTA2 (SMA) were analyzed by RT-PCR. Values are represented in log2 scale and show the fold change versus WT. Data are mean ± SEM ($n = 3$) and Tie-1($n = 4$). **e** WT MSCs were incubated in 0.5% FBS and pre-treated with the p38α inhibitors SB203580 (SB), PH787904 (PH) and BIRB0796 (BIRB). The following day, $1 \times 10^4$ cells were seeded in matrigel with 0.5% FBS and the inhibitors, and tube formation was quantified 16 h later. The histogram shows the quantification of the branches length per field using ImageJ. Data are mean ± SEM ($n = 5$ in p38αΔ, $n = 4$ in WT, WT + PH and WT + Birb, $n = 3$ in WT + SB). Bars, 100 μm

JNK hyperactivation in TGF-β-treated p38αΔ MSCs (Fig. 6b), suggesting that defective negative regulation of TAK1 by p38α accounts for the observed increase in JNK activity.

JNK signaling plays an important role in the stress response[29], but this pathway has been also linked to endothelial cell proliferation and VEGF expression[16,31]. We tested the contribution of JNK signaling to the TGF-β-induced angiogenic program and capacity to form capillary-like structures observed in p38αΔ MSCs. We found that JNK inhibition impaired the TGF-β-induced phosphorylation of Ser-208 in the Smad3 linker (Fig. 6c), and decreased angiogenic gene expression in p38αΔ MSCs, both under basal conditions and in response to TGF-β (Fig. 6d and Supplementary Fig. 4c). Moreover, chemical inhibition or siRNA-mediated downregulation of JNK1 and JNK2 (Supplementary Fig. 4d) impaired capillary tube formation in matrigel by p38αΔ MSCs both under basal conditions and after TGF-β stimulation (Fig. 6e and Supplementary Fig. 4e). These results support a role for JNK in the mesenchymal to endothelial conversion of MSCs induced by TGF-β.

**Mesenchymal p38α negatively regulates tumor angiogenesis.** Next, we analyzed the contribution of p38α in MSCs to tumorigenesis. We found that injection of nude mice with p38αΔ MSCs and HT29 colorectal cancer cells reduced the latency and increased the size of the tumors formed, in comparison with HT29 cells injected alone or with WT MSCs (Fig. 7a, b). Similar results were observed with CMT93 cancer cells (Supplementary Fig. 5a, b). We analyzed the tumors formed by HT29 cells, and observed that co-injection with p38αΔ MSCs increased cell proliferation but did not affect apoptosis in comparison with WT MSCs (Fig. 7c). Moreover, tumors formed by HT29 cells and p38αΔ MSCs showed elevated levels of phospho-Smad2 in tumor-associated stromal cells, indicating the induction of a stromal TGF-β response, which was not observed in tumors formed by HT29 cells alone or with WT MSCs (Fig. 6d). Of note, HT29 cells have homozygous *SMAD4* mutations and do not respond to TGF-β, yet this alteration does not prevent upstream signaling and nuclear accumulation of phospho-Smad2[32]. The increased TGF-β signaling in tumors with p38αΔ MSCs was accompanied by enhanced angiogenesis, while the number of SMA⁺ cells was reduced (Fig. 7e, f). These results suggest that tumors with p38αΔ MSCs are more prone to induce blood vessel formation but showed decreased expression of fibroblast markers, in agreement with the mRNA expression data (see above, Fig. 3d). Consistent with this idea, the conditioned media from HT29 cancer cells increased capillary formation and CD31 expression in p38αΔ but not WT MSCs (Supplementary Fig. 5c, d), whereas SMA expression was lower in p38αΔ MSCs (Supplementary Fig. 5e).

**Detection of perivascular MSCs in mouse colon.** Given the importance of angiogenesis for inflammation, we choose a model of inflammation-associated colon tumorigenesis to investigate the function of p38α in the mesenchymal to endothelial transition in vivo.

We generated p38αΔ^FSP1 mice bearing *Mapk14*^lox/lox alleles and the Fibroblast-specific protein 1 (FSP1)-Cre transgene, which is active in a large subset of fibroblasts[33]. As a control, we used FSP1-Cre mice that are WT for p38α. These mice also contained a Cre-inducible Tomato-GFP reporter transgene[34], so that cells normally express Tomato but upon FSP1-Cre expression the Tomato cassette is excised allowing the irreversible expression of the GFP that marks FSP1⁺ cells and their progeny.

Perivascular cells displaying MSC characteristics have been identified based on the expression of CD146 (M-CAM) and PDGFRB[2,3]. We observed that PDGFRB⁺ GFP⁺ cells sorted from normal colon mucosa of WT and p38αΔ^FSP1/Tomato-GFP mice represented about 66% of the total GFP⁺ population (Supplementary Fig. 6a). Moreover, PDGFRB⁺ GFP⁺ cells showed substantial deletion of the floxed exon2 of *Mapk14* confirming that FSP1-Cre induces p38α downregulation in the mesenchymal cell compartment (Supplementary Fig. 6b).

To ascertain that FSP1-Cre was expressed in pericyte-like cells, we stained colon sections from FSP1-Cre/Tomato-GFP mice both with GFP and with antibodies to CD31 to identify endothelial cells. We found GFP⁺ cells located around the blood vessels in the perivascular zone of large arteries, arterioles and microvascular capillaries from normal colon tissue (Fig. 8a), suggesting a MSC-like identity. Consistent with this idea, PDGFRB and CD146 co-expressed with GFP in perivascular locations around the tunica media layer of large arteries and arterioles labeled with CD31 (Supplementary Fig. 6c). Analysis by FACs showed that about 48% of the GFP⁺ cells in colons from FSP1-Cre/Tomato-GFP mice expressed PDGFRB and CD146 (Supplementary Fig. 6d). These results indicate that GFP⁺ (i.e., FSP1 expressing) cells from colon tissue express perivascular markers in vivo and are located in the vascular wall. FSP1 is considered a marker of mesenchymal cells in different organs undergoing tissue remodeling and has been used to identify fibroblasts, however FSP1⁺ cells have been reported to express F4/80 and other markers of the myeloid-monocytic linage[35]. We found that only about 2% of FSP1-GFP⁺ cells co-expressed CD31 and CD45 in basal conditions, and no co-expression of CD68 (macrophage) and endothelial markers was found in blood vessel cells. These results indicate that, under our experimental conditions, FSP1 is barely expressed in macrophages, and also show that macrophages are not incorporated in the colon tissue vasculature (Supplementary Fig. 6e, f).

Furthermore, we sorted PDGFRB⁺ and CD146⁺ cells from colon tissue of UBC-Cre-ERT2 mice (Supplementary Fig. 6g),

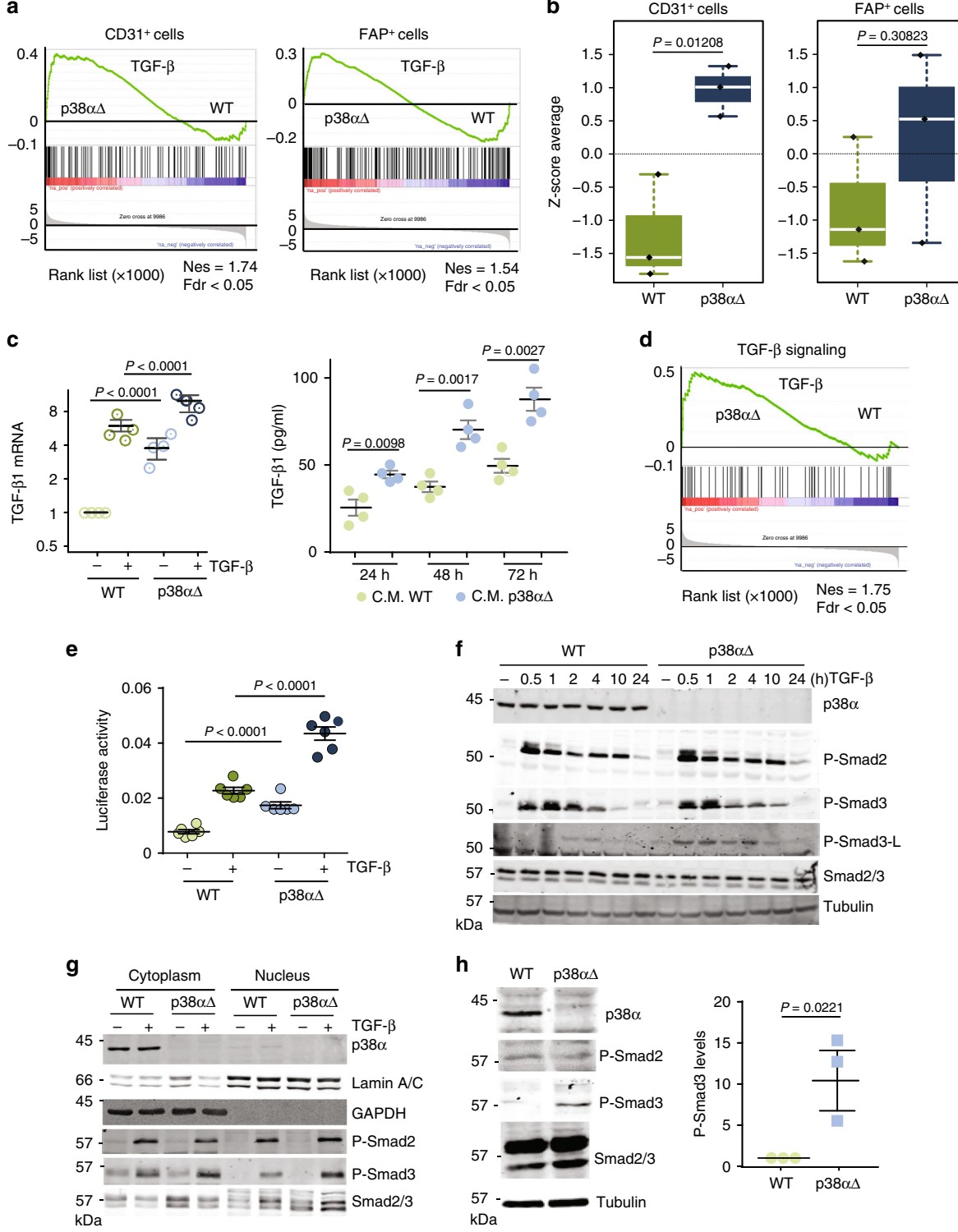

which were then cultured and treated with 4-OHT to delete p38α (p38αΔ). Treatment with TGF-β induced Smad2 phosphorylation to similar extents in WT and p38αΔ perivascular cells (Supplementary Fig. 6f). Strikingly, we found that p38αΔ but not WT perivascular cells were able to form tube-like structures in matrigel, which was enhanced by TGF-β treatment (Fig. 8b). Taken together, these results suggest a similar role for p38α in the regulation of TGF-β -induced capillary tube formation ex vivo in both perivascular cells from colon and MSCs.

**p38α-null mesenchymal cells adopt endothelial fate in vivo.** MSCs in the perivascular niche may become activated upon organ injury and support tissue regeneration[36]. Therefore, we investigated the contribution of MSCs to vasculature formation in a model of inflammation-associated colon tumorigenesis.

To address the implication of p38α in the mesenchymal to endothelial cell differentiation in vivo, p38αΔ[FSP1] mice were treated with AOM/DSS to induce colon tumors. In response to DSS, p38αΔ[FSP1] mice showed enhanced body weight loss

**Fig. 4** p38α controls TGF-β signaling pathway. **a** Comparison of TGF-β treated WT and p38αΔ MSCs according to gene expression signatures of endothelial cells (CD31[+]) and CAFs (FAP[+]) isolated from human primary CRC tumors. Values are z-scores with standard deviation (SD) from the mean (*n* = 3). **b** Gene set enrichment analysis (GSEA) plots showing enrichment of gene expression signatures of endothelial cells (CD31[+]) and CAFs (FAP[+]) isolated from human primary colorectal tumors in transcriptomic programs induced by TGF-β in WT and p38αΔ MSCs. Nes, normalized enrichment score; Fdr, false discovery rate. **c** (right) Relative TGF-β1 mRNA levels in MSCs treated as indicated were analyzed by RT-PCR. Values are represented in log2 scale and show the fold change versus WT. Data are mean ± SEM (*n* = 4). (left) TGF-β1 protein levels were determined by ELISA in conditioned media (C.M.) from WT and p38αΔ MSCs after 24, 48 and 72 h of culture Data are mean ± SEM (*n* = 4). **d** GSEA identified TGF-β Signaling (M5896) as one of the main processes affected in TGF-β-treated p38αΔ MSCs compared with WT MSCs. Nes, normalized enrichment score; Fdr, false discovery rate. **e** MSCs were co-transfected with the reporter under control of Smad-binding elements (CAGA-LUC) and RNL-TK as a control, and 32 h later were treated with TGF-β for 4 h. Luc activity was normalized to the activity of RNL-TK. Data are mean ± SEM (*n* = 3). **f** MSCs were treated with TGF-β and collected at the indicated times followed by immunoblotting analysis. The indicated molecular weights for the P-Smad2, P-Smad3 and P-Smad3-L immunoblots represent pre-stained markers. **g** Nuclear and cytoplasmic fractions from MSCs either untreated or treated with TGF-β for 20 h were analyzed by immunoblotting with the indicated antibodies, using GAPDH and Lamin A/C as cytoplasmic and nuclear markers, respectively. **h** Total lysates (80 μg of protein) from non-treated WT and p38αΔ MSCs were analyzed by immunoblotting. The histogram shows the quantification of phospho-Smad3 levels in three independent experiment using ImageJ. Values show the fold change versus WT. Data are mean ± SEM (*n* = 3)

compared with WT littermates, suggesting enhanced epithelial damage in mice with p38α downregulation in FSP1[+] cells (Supplementary Fig. 7a). At the end of the treatment, p38αΔ[FSP1] mice showed more macroscopic colon tumors (Supplementary Fig. 7b), as well as increased average tumor size and more tumors larger than 4 mm (Fig. 8c). Moreover, tumors from p38αΔ[FSP1] mice showed increased staining with CD31 antibodies (Fig. 8d and Supplementary Fig. 7c). We also found increased levels of phosphorylated Smad2 in the tumor stroma of p38αΔ[FSP1] mice (Supplementary Fig. 7d), consistent with a local upregulation of TGF-β signaling in stromal cells. Taken together, these data show that p38α downregulation in the mesenchymal cell compartment leads to increased angiogenesis and colon tumor formation.

To test whether p38αΔ MSCs undergo mesenchymal-to-endothelial transition during tumorigenesis in vivo, we analyzed the expression of endothelial markers in FSP1-Cre/Tomato-GFP mice. In untreated mice, we detected GFP[+] mesenchymal cells expressing CD31 or Ve-Cadherin in the blood vessel from colon of p38αΔ[FSP1] mice whereas in WT mice GFP was expressed in perivascular areas (Fig. 8a and Supplementary Fig. 7e). Analysis by FACS showed that colon tissue from p38αΔ[FSP1] mice contained a higher proportion of GFP[+] cells expressing CD31 either alone (Fig. 8e and Supplementary Fig. 7e) or together with PDGFRB (Fig. 8f and Supplementary Fig. 7f). Moreover, we estimated that 11% of endothelial cells from p38αΔ[FSP1] mice expressed GFP (Supplementary Fig. 7g).

We also investigated the co-expression of GFP with endothelial markers in FSP1-Cre/Tomato-GFP mice treated with AOM/DSS to induce colon tumors. Blood vessels co-expressing GFP and CD31, Ve-Cadherin, eNOS or CD105[37] were observed in the colon tumors from p38αΔ[FSP1] but not WT mice (Fig. 8g and Supplementary Fig. 8a, b). These results were confirmed by FACS analysis, which also showed a high proportion of GFP[+] cells expressing CD31 either alone or together with PDGFRB in colon tumors from p38αΔ[FSP1] mice (Fig. 8e, f). We also used mice expressing PDGFRB-Cre-ERT2[38] and the Tomato-GFP reporter as an alternative model to confirm that p38α regulates the mesenchymal to endothelial cell differentiation in vivo. For these experiments, adult mice (p38α WT) were treated with 4-OHT to induce Cre activation, and then DSS was administered to induce epithelial damage in the presence or absence of the p38α inhibitor PH797804 (Supplementary Fig. 8c). Activation of Cre in the perivascular zone was confirmed by staining colon sections from untreated PDGFRB-Cre-ERT2/Tomato-GFP mice with both GFP and CD31 antibodies (Supplementary Fig. 8d). FACs analysis showed that in colons from non-treated mice, only about 1% of the GFP[+] cells co-expressed CD31 and CD146 (Fig. 8h). Treatment with PH797804 slightly but significantly increased

the number of triple positive cells (GFP[+] CD31[+] CD146[+]) detected by FACs in untreated colons (Fig. 8h). Moreover, we found more GFP[+] perivascular cells expressing CD31 in the blood vessels of colons from PH797804-treated mice (Supplementary Fig. 8d). Interestingly, treatment of mice with DSS substantially increased the number of GFP[+] cells expressing CD31 and CD146 (Fig. 8h and Supplementary Fig. 8e). The proportion of GFP[+] CD31[+] CD146[+] cells further increased in colons from mice treated with both DSS and PH797804, which correlated with more GFP[+] CD31[+] cells incorporated into the vessels of damage areas (Supplementary Fig. 8f). These results support that mesenchymal cells can switch into endothelial-like cells when the colon is damaged, and this process is potentiated by p38α inhibition.

## Discussion

In the early stages of colorectal tumorigenesis, the tumor stroma shows increased expression of pro-angiogenic factors and downregulation of anti-angiogenic factors, which allow the initiation of neovascularization. We have found that p38α signaling controls the angiogenic switch by negatively regulating new vascularization in mouse models and PDXs of colon tumorigenesis. These observations are aligned with previous results showing that excessive p38α activation reduces pathological neoangiogenesis in the retina of mice with endothelial-specific Atm deficiency[15].

The vascular cells in adult blood vessels were considered to be terminally differentiated and relatively quiescent, but several investigations have identified a vasculature reservoir of potential progenitor cells. These include perivascular cells expressing MSC markers and retaining multi-lineage potential ex vivo[2,3]. However, it has been recently suggested that the plasticity of endogenous pericytes observed ex vivo or following transplantation in vivo arises from the manipulation of cells[4]. Our results show that MSCs induce angiogenesis not only by promoting migration and invasion of endothelial cells but also by acquiring endothelial cell characteristics, and these processes are negatively regulated by p38α signaling. Moreover, silencing p38α enhances the co-expression of perivascular and endothelial markers in tumor vasculature.

TGF-β has an important pro-tumoral effect in human colorectal tumors[25,39]. The induction of angiogenesis is key for tumor progression, and several studies have illustrated pro-angiogenic effects of TGF-β[40,41]. Interestingly, expression profiling reveals that TGF-β-treated p38αΔ MSCs display enhanced expression of angiogenic genes, and acquire features of endothelial cells in human colorectal tumors. We show that the acquisition of endothelial characteristics observed in p38αΔ MSCs correlates

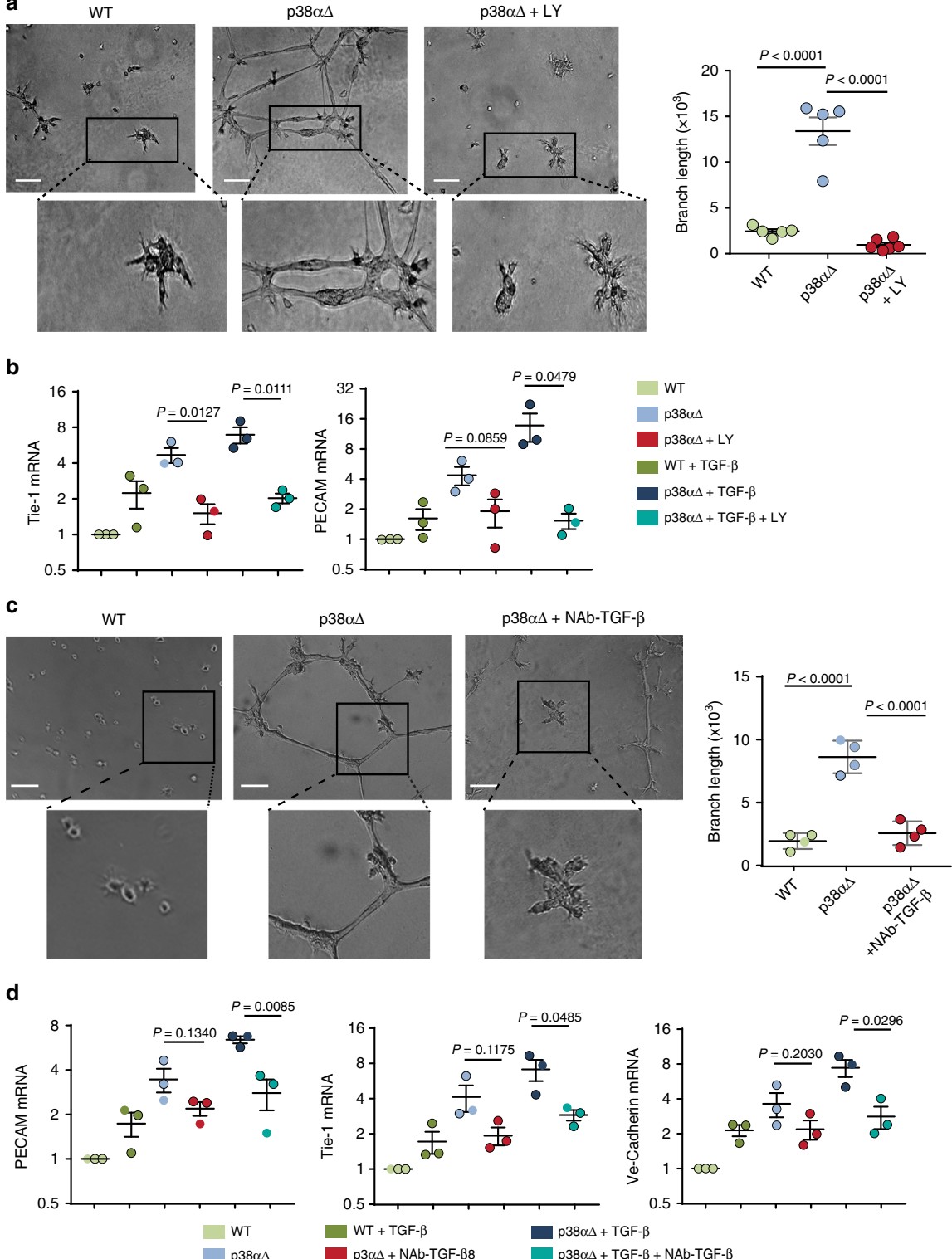

**Fig. 5** TGF-β signaling controls the conversion of p38α-deficient MSCs to endothelial cells. **a** MSCs were seeded in 0.5% FBS, and treated with the TGF-β inhibitor LY2157299 (LY, 1 nM) as indicated. Tube formation in matrigel was evaluated 10 h later. The histogram shows the quantification of the branch length per field using ImageJ. Data are mean ± SEM ($n = 5$ in WT and p38αΔ, $n = 6$ in p38αΔ + LY). Bars, 100 μm. **b** Cells were cultured in 0.5% FBS for 5 days in the presence or absence of TGF-β and/or the TGF-β inhibitor LY, which was added to the medium every 2 days. Relative mRNA expression levels for the endothelial markers Tie-1 and PECAM were analyzed by RT-PCR. Values are graphically represented in log2 scale and show the fold change versus WT. Data are mean ± SEM ($n = 3$). **c** MSCs were treated with TGF-β 1, 2, 3 antibody (2 μg ml$^{-1}$) or mouse IgG1, as indicated. Tube formation in matrigel was evaluated 10 h later. The histogram shows the quantification of the branch length per field using ImageJ. Data are mean ± SEM ($n = 4$) Bars, 100 μm. **d** Cells were cultured in 0.5% FBS for 5 days in the presence or absence of TGF-β and/or the TGF-β 1, 2, 3 Antibody or mouse IgG1, which was added to the medium every 2 days. Relative mRNA expression levels for the endothelial markers Tie-1, PECAM and Ve-Cadherin were analyzed by RT-PCR. Values are graphically represented in log2 scale and show the fold change versus WT. Data are mean ± SEM ($n = 3$)

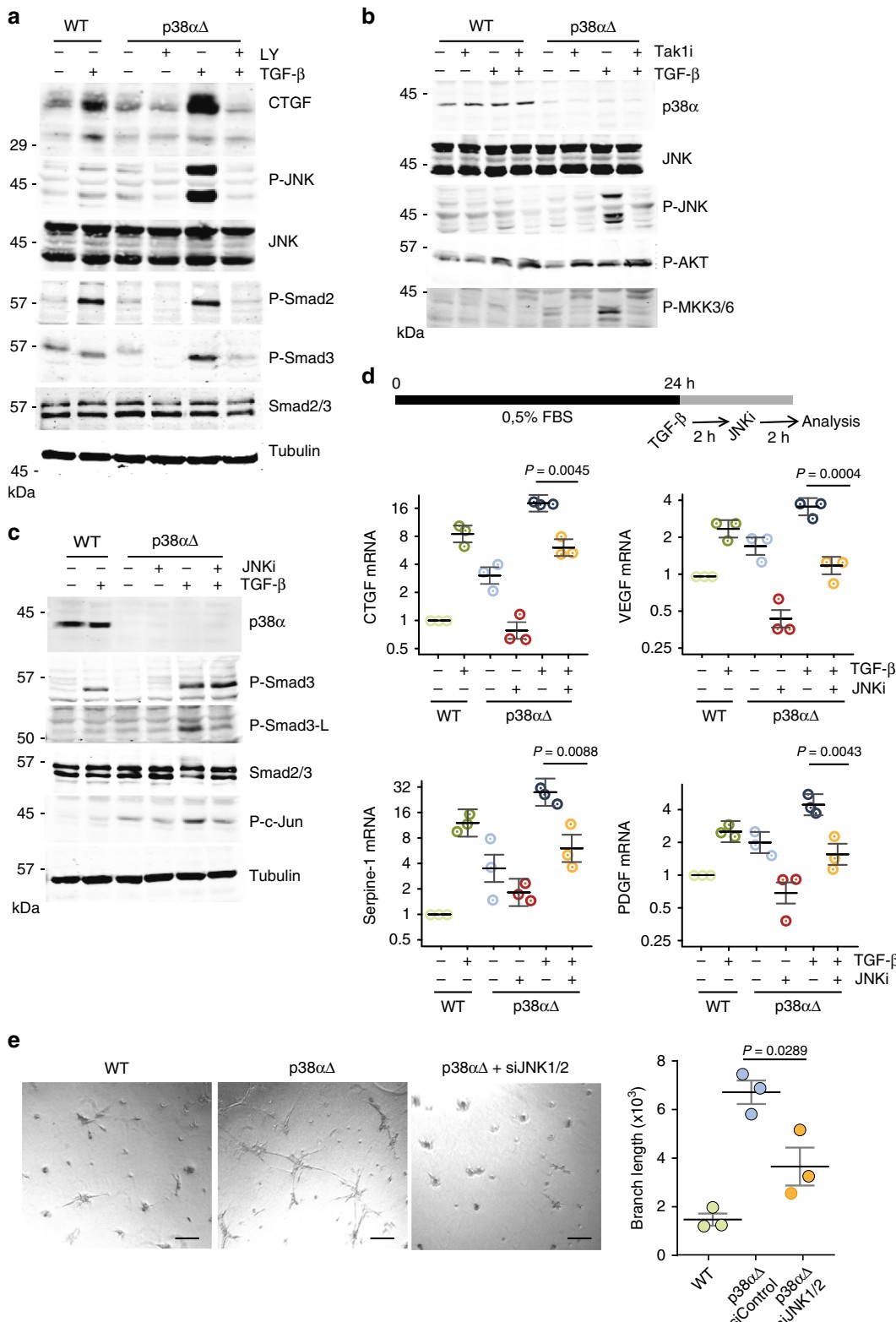

with TGF-β release and increased Smad3 phosphorylation, which can be impaired by ALK5 receptor inhibition or by a TGF-β neutralizing antibody. Overall, our results suggest that TGF-β induces the commitment of MSCs to endothelial cells through the ALK5 receptor, and illustrates a feedback loop by which p38α negatively regulates ALK5 signaling.

Phosphorylation of the Smad C-terminus is key for activation, but additional phosphorylations in the linker region can also positively and negatively regulate Smad function. Interestingly, p38αΔ MSCs showed enhanced phosphorylation of Ser-208 in the Smad3 linker, which can be phosphorylated by JNK. Phosphorylation of the Smad3 linker has been reported to facilitate nuclear translocation, transcriptional activation and interaction with coactivators[42,43]. Previous reports have described the crosstalk between the p38α and JNK pathways at several levels[29,44]. We show that p38α negatively regulates JNK activation

**Fig. 6** p38α regulates the expression of TGF-β target genes though the JNK pathway. **a** Starved MSCs were pre-treated with the TGF-β inhibitor LY 2157299 (LY, 1 nM) for 2 h and then with TGF-β for 20 h, and total protein lysates were analyzed by immunoblotting with the indicated antibodies. **b** MSCs were cultured in 0.5% FBS for 24 h, pre-treated with the TAK1 inhibitor (5Z)-7-Oxozeaenol (TAKi, 1 μM) for 2 h and then treated with TGF-β for 4 h. Total cell lysates were analyzed by immunoblotting with the indicated antibodies. **c** MSCs were cultured in 0.5% FBS for 24 h, pre-treated with the JNK inhibitor BI-78D3 (JNKi, 2 μM) for 2 h and then treated with TGF-β for 4 h more. Total cell lysates were analyzed by immunoblotting with the indicated antibodies. **d**, MSCs were cultured in 0.5% FBS for 24 h, and then TGF-β and/or BI-78D3 (JNKi, 2 μM) were added to the medium sequentially with a 2 h interval, as shown in the scheme. Relative levels for the indicated mRNAs were determined by qRT-PCR. Values are graphically represented in log2 scale and show the fold change versus WT. Data are mean ± SEM ($n = 3$). **e** WT and p38αΔ MSCs, as well as p38αΔ treated with JNK1 and JNK2 siRNAs were incubated in matrigel with 0.5% of FBS and tube formation was analyzed 20 h later. The histogram shows the average branch length per field in each case. Quantifications were performed using ImageJ. Data are mean ± SEM ($n = 3$). Bars, 100 μm

---

in MSCs, probably by inhibiting TAK1. The balance between JNK and p38α signaling determines cell fate in response to various stress stimuli[45,46], and JNK mediates angiogenesis by regulating VEGF[17]. Therefore, we hypothesize that the enhanced TGF-β and JNK-dependent angiogenic program observed in MSCs could be mediated by the cooperation between AP-1 and Smad3/Smad4 transcriptional activities[13,47].

TGF-β upregulation correlates with poor prognosis, increased tumor growth and angiogenesis in several cancers. Conversely, TGF-β signaling antagonists reduce tumor growth and metastasis[10,39]. We show that p38α signaling in mesenchymal cells negatively regulates tumor growth, angiogenesis and the TGF-β response both in colon cancer xenografts and mouse models. The enhanced angiogenesis induced by p38α inhibition correlates with the commitment of MSCs to an endothelial phenotype rather than to the myofibroblasts that normally originate from MSCs in perivascular locations[3,5]. As a consequence, p38α inhibition leads to the downregulation of fibroblast markers with concomitant upregulation of pro-angiogenic factors.

FSP1 was originally proposed as fibroblast-specific marker[33]. Using a Tomato-GFP reporter, we show that FSP1$^+$ cells in the colon are located in the perivascular niche and express MSC markers. Our genetic analysis using p38α$^{FSP1}$/Tomato-GFP mice indicates that p38α downregulation induces the formation of new vasculature and stimulates the transdifferentiation of resident PDGFRB$^+$FSP1$^+$ cells into endothelial-like cells during colon tumorigenesis in vivo. Since myeloid cells and lymphocytes infiltrating injured tissues can express FSP1[35], we used mice PDGFRB-Cre-ERT2/Tomato-GFP to confirm that mesenchymal cells acquire an endothelial-like phenotype in response to DSS-induced intestinal damage. In these experiments, PDGFRB-Cre-ERT2 was activated in perivascular cells of adult mice, avoiding the possible labelling of other cell populations during embryogenesis, as reported in the developing heart[38], which could confound the interpretation of results. Furthermore, p38α negatively regulates the vasculogenic properties of PDGFRB$^+$CD146$^+$ perivascular cells from mouse colon ex vivo. Overall, our results indicate that mesenchymal cells can differentiate into endothelial cells during colon tissue repair and tumorigenesis. The mesenchymal-to-endothelial transition has been proposed to occur in cardiac fibroblasts contributing to neovascularization of the injured heart[48], but this was later challenged using different tools[49]. It is worth mentioning that in our study p38α downregulation plays a key role in the mesenchymal to endothelial conversion.

Diverse progenitor cells probably exist in adult wall vasculature, and may partly account for the phenotypic and functional heterogeneity observed in homeostasis and disease conditions[1]. Moreover, it has been reported that isolated Tbx18$^+$ pericytes have the potential to differentiate ex vivo but not in vivo[4]. These observations highlight the importance of the endogenous niche to regulate the differentiation potential of progenitor cells, suggesting the existence of specific signaling pathways/gene regulation

programs that modulate cell fate decision of perivascular progenitors. Our results indicate a key role for p38α restraining the angiogenic program in mesenchymal cells, including their conversion to an endothelial-like phenotype. This p38α function is mediated by the downregulation of TGF-β and JNK signaling, which are both required for these processes. Thus, TGF-β produced by the tumor microenvironment may control the angiogenic switch through the regulation of MSC function, which triggers angiogenesis by promoting migration and invasion of endothelial cells, as well as by providing a novel source of endothelial cells.

In summary, the interplay between p38α and TGF-β/JNK signaling emerges as a new mechanism that regulates the response to angiogenic cues and cell fate decision of mesenchymal/perivascular cells in injured tissues and tumors (Fig. 9). We propose that changes in p38α activity, which could be induced by signals from the niche, may in turn regulate the TGF-β-induced pro-angiogenic response through the JNK pathway. Recruitment and proliferation of endothelial cells are critical for tumor growth, and our findings highlight the importance of p38α signaling modulation to control not only the production of pro-angiogenic factors that enhance tumor angiogenesis but also the incorporation of mesenchymal-derived endothelial cells into the tumor vasculature, both mechanisms are intrinsically associated and are likely to contribute to angiogenesis in colon tumors as well as in damage tissue.

## Methods

**Mice.** The floxed allele of *Mapk14* encoding p38α[50,51] was combined with the UBC-Cre-ERT2[52], FSP1-Cre[33], PDGFRB-Cre-ERT2[38] and with the lineage reporter Tomato/GFP inserted into the Rosa26 locus[34]. Mice used for experiments were maintained in C57BL/6 J background and were homozygous for *Mapk14*$^{lox}$ and Tomato/GFP, and heterozygous for the different Cre lines. Control mice were homozygous for Tomato/GFP and heterozygous for Cre. Mice were housed according to national and European Union regulations, and protocols were approved by the animal care and use committee of Barcelona Science Park (CEEA-PCB). Genotyping was performed by PCR using 25 ng of tail genomic DNA and the following PCR conditions 94 °C for 5 min, 94 °C for 30 s, 55 °C for 45 s, 72 °C for 30 s, step 2-4 for 35 cycles, 72 °C for 10 min, and 4 °C forever. PCR products were resolved by agarose gel electrophoresis. The primers used were for Cre (amplicon size 520 bp) Fw: ACGAGTGATGAGGTTCGCAAG and Rv: CCCACC GTCAGTACGTGAGAT, and for *Mapk14* amplicon sizes are 121 bp for the WT and 188 bp for the floxed alleles) Fw: ATGCTACTGTCTGCGCCTCTCT and Rv: CAGCTTCTTAACTGCCACACGA and for Tomato-GFP (amplicon size 284 for mT/mG and 297 for WT) Fw: CTCTGCTGCCTCCTGGCTTCT; Rv: CGAGGC GGATCACAAGCAATA Rv: TCAATGGGCGGGGGTCGTT.

**Induction of tumors in mice.** Colitis-associated colon tumors were induced as previously described[19]. Briefly, 8-week-old mice were injected i.p. with AOM (10 mg kg$^{-1}$ per mice, Sigma # A5486). Five days after the AOM injection, mice were fed for 5 days with 2% DSS (w/v) (MP Biochemicals 216011090 MW, 36,000–50,000 Da) ad libitum in the drinking water, which was followed by 14 days of normal drinking water. The DSS treatment was repeated for two additional cycles. On day 100, the colon was removed, flushed with PBS and opened longitudinally. Macroscopically visible tumors were counted and measured. Colon sections were fixed in 10% formalin and paraffin embedded for immunohistochemical analysis, or directly frozen for immunoblot and RNA analysis.

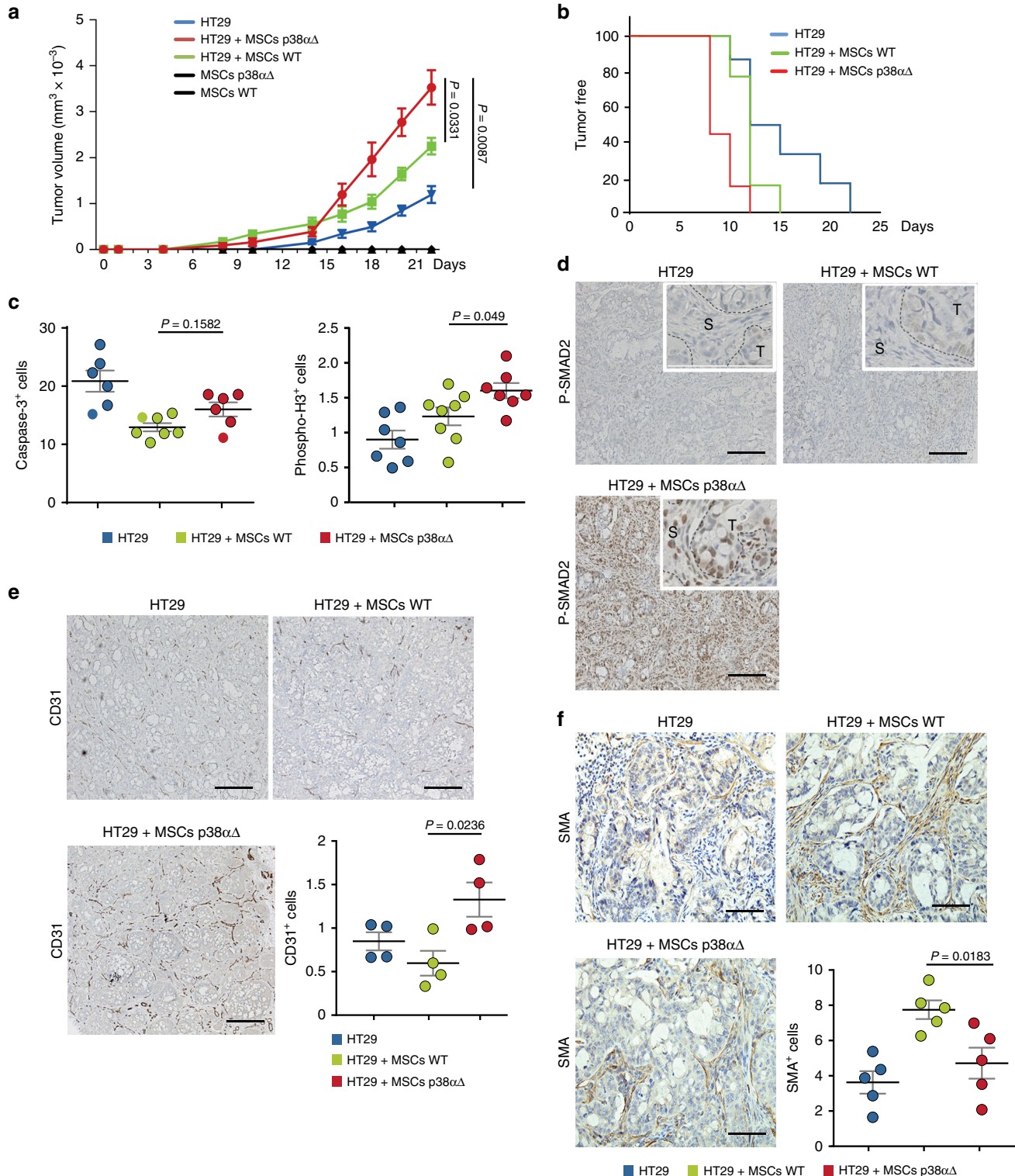

**Fig. 7** p38α-deficient MSCs enhance tumor formation and angiogenesis by colorectal cancer cells. **a** Growth kinetics of $2 \times 10^6$ HT29 colon cancer cells subcutaneously implanted into nude mice either alone or together with $5 \times 10^5$ WT or p38αΔ MSCs. As a control, MSCs were also implanted alone. Values are mean ± SEM ($n = 8$ tumors). **b**, Kaplan-Meier curves for the mice in (**a**). **c** Quantification of the number of Caspase-3+ cells per field and of the area covered by phospho-Histone 3+ cells (percentage per field) as markers of apoptosis and proliferation, respectively, in tumors obtained as in (**a**). Data are mean ± SEM ($n = 6$ or 7 tumors). **d** Phospho-Smad2 antibody staining of tumors obtained as in **a**. T, tumor; S, stroma. Bars, 100 μm. **e** CD31 antibody staining of tumors obtained as in (**a**). Bars, 100 μm. The percentage of CD31+ cells among the total number of cells per field was quantified using ImageJ. Data are mean ± SEM ($n = 4$ tumors). **f** SMA antibody staining of tumors obtained as in (**a**). Bars, 100 μm. The number of SMA+ cells per field was quantified using ImageJ. Data are mean ± SEM ($n = 5$)

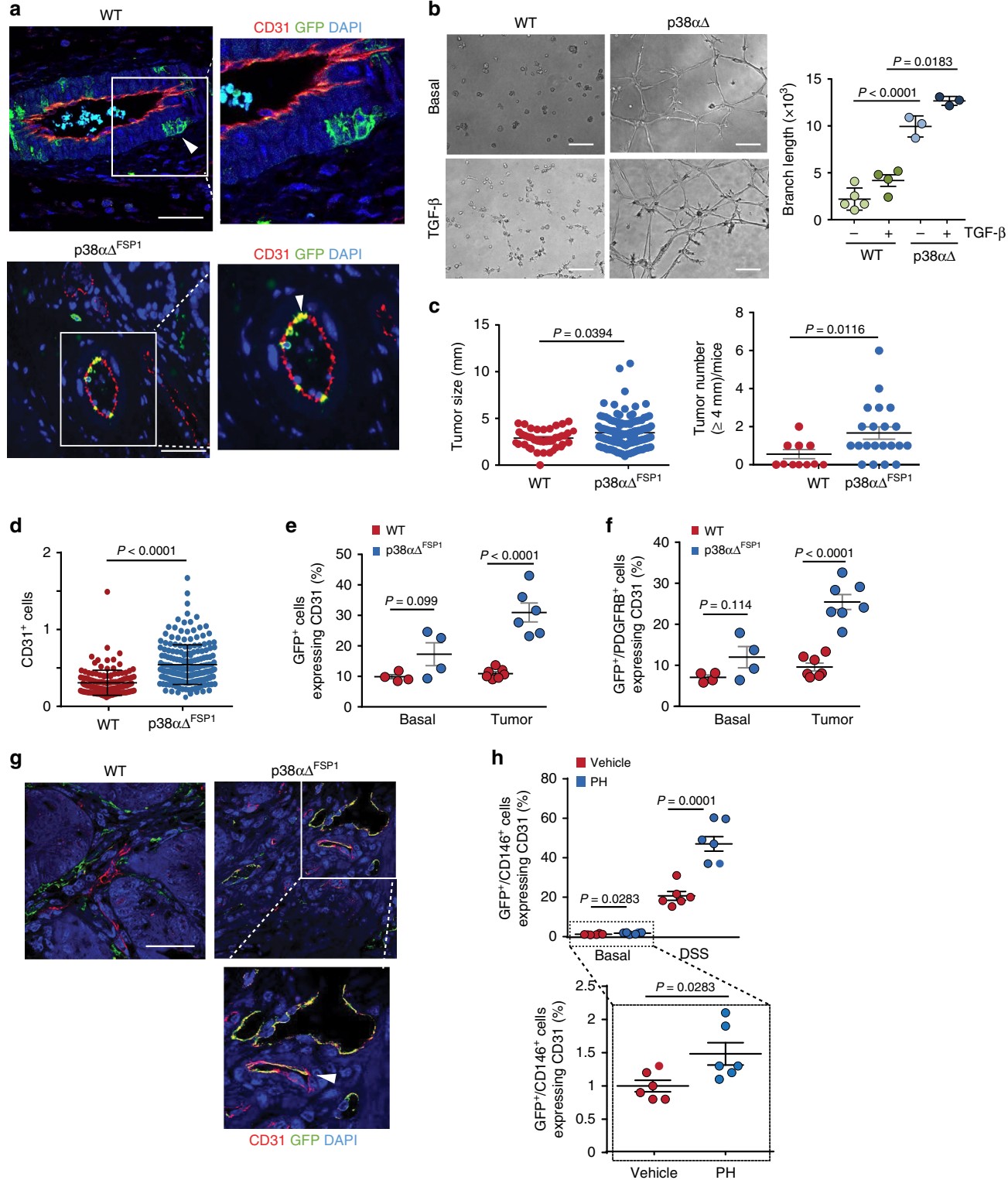

For tumor xenografts, $2 \times 10^6$ HT29 cells (ATCC HTB-38™) or CMT93 cells (ATCC CCL-223™), previously tested for mycoplasm contamination, were combined with $5 \times 10^5$ WT or p38α∆ MSCs and subcutaneously injected into the flanks of 8-week-old athymic nude mice (Harlan). Tumor growth was followed every other day after the first week. Mice were sacrificed by cervical dislocation when the group of faster growing tumors reached about 3 cm$^3$ (calculated by the ellipsoid volume, usually 4 to 10 weeks after the injection, depending on the cell line used), the tumor area was collected, fixed in formalin, and embedded in paraffin.

**Induction of colitis in mice**. Adult p38α WT mice (8 weeks old) were treated with 4-OHT for 5 days to induce PDGFRB-Cre-ERT2 activation and expression of the

GFP reporter, and 9 days later started to be treated daily with the p38α inhibitor PH797804 or vehicle until the end of the experiment (Supplementary Fig. 8c). Four days after initiation of the PH797804 treatment, DSS was administered ad libitum in the drinking water for 5 days. Mice were sacrificed two days after finishing the DSS treatment, the colon was removed, flushed with PBS and either opened longitudinally, fixed in 10% formalin and paraffin embedded for immunohistochemical analysis, or directly processed for FACs analysis as mentioned below.

**MSC culture, differentiation assays, and treatments**. Murine MSCs were obtained as indicated[53] from *Mapk14*$^{lox/lox}$;UBC-Cre-ERT2 mice and were cultured in Dulbecco-modified Eagle's medium (DMEM) plus 10% fetal bovine serum

**Fig. 8** p38α regulates colon tumor angiogenesis and the expression of endothelial markers by PDGFRB+ mesenchymal cells. **a** Colons from non-treated WT and p38αΔ^FSP1 Tomato-GFP mice were immunostained with antibodies for GFP (green) and CD31 (red). Co-staining is indicated by white arrowheads. Bars, 100 μm. The right panels show higher magnifications. **b** PDGFRB+/CD146+ colon perivascular cells were seeded in matrigel in the presence or absence of TGF-β and tube formation was determined 16 h later. The histogram shows the quantification of branch lengths. Bars, 100 μm. Data are mean ± SEM (n = 4 WT, n = 3 p38αΔ). **c** Tumor sizes and number of tumors larger than 4 mm formed in WT and p38αΔ^FSP1 mice treated with AOM/DSS. Data represent mean ± SEM (n = 11 WT, n = 21 p38αΔ^FSP1 mice). **d** Colon sections from WT and p38αΔ^FSP1 Tomato-GFP mice were stained for CD31. Bars, 100 μm. The percentage of CD31+ cells in the total number of cells per tumor area was quantified on pictures taken from colon tumors. Data represent mean ± SEM (n = 8 WT, n = 9 p38αΔ^FSP1 mice). **e** Quantification by FACS of the percentage of GFP+ cells expressing CD31 in normal colons (Basal) and in tumors. Data are mean ± SEM (n = 4 Basal, n = 7 WT, n = 6 p38αΔ^FSP1 tumors). **f** Quantification by FACS of the percentage of cells expressing both GFP and PDGFRB that also express CD31 in normal colons (Basal) and colon tumors from WT and p38αΔ^FSP1 Tomato-GFP mice. Data are mean ± SEM (n = 4 Basal, n = 7 Tumor). **g** Colon tumors from WT and p38αΔ^FSP1 Tomato-GFP mice were immunostained for GFP and CD31. Co-staining is indicated by a white arrowhead. Bars, 100 μm. The right panel shows a higher magnification of the indicated area. **h** Quantification by FACS of the percentage of colon cells expressing GFP and CD146 that also express CD31. PDGFRB-Cre-ERT2/ Tomato-GFP mice were treated with 4-OHT and then either with the p38α inhibitor PH797804 (PH) or vehicle followed by DSS for 5 days, as indicated. Data are mean ± SEM (n = 6)

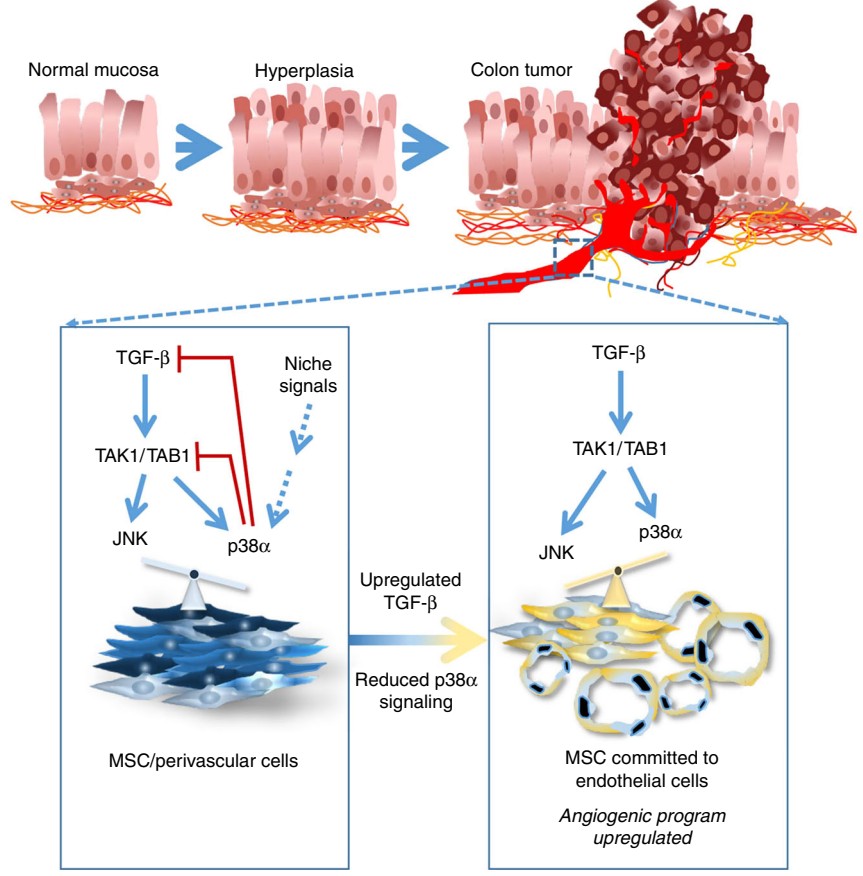

**Fig. 9** Schematic illustration of the interplay between p38α and TGF-β/JNK signaling that regulates angiogenic cues and cell fate decision of mesenchymal/ perivascular cells in injured tissues and tumors. See main text for details

(FBS). Immortalization was performed using a standard 3T3 protocol, and immortalized cells were tested for mycoplasm contamination weekly. Surface markers of MSCs, endothelial cells and leukocytes were analyzed by flow cytometry using the antibodies indicated in Supplementary Table 1. WT MSCs were incubated with 5% FBS in PBS for 30 min at RT, followed by incubation with the indicated antibodies on ice, and washed with PBS + 5% FBS. Samples (10,000 events) were analyzed using a FACS Aria 2.0 (BD Biosciences) flow cytometer and the FlowJo software.

To induce deletion *Mapk14* deletion and the downregulation of p38α, MSCs were treated for 2 days with 1 μM of 4-OHT (Sigma #H7904). When indicated, cells with treated with TGF-β (5 ng/ml, Peprotech #100-21), the p38 MAPK inhibitors PH797804 (1 μM, Selleckchem #S2726), BIRB0796 (200 nM, Axon Medchem #AX1358) and SB203580 (10 μM, Axon Medchem #AX1363), the TAK1 inhibitor (5Z)-7-Oxozeaenol (1 μM), the JNK inhibitors BI-78D3 (2 μM, Sigma #B3063) and SP600126 (20 μM, Calbiochem # 420119), or the TGF-β inhibitor LY 2157299 (1 nM, provided by Eduard Batlle)[54].

Osteogenic differentiation of MSCs was induced by adding 1 μM dexamethasone (Sigma #D4902), 10 mM β-glycerophosphate (Sigma #G9422) and 50 μM L-ascorbic acid-2-phosphate (Sigma #A4544) to a confluent culture for 3 days. Cells were cultured in DMEM plus 10% FBS for 10 additional days and stained with Alizarin Red (Sigma #A5533). For adipogenic differentiation, cells were incubated with 1 μM dexamethasone, 0.1 mg/ml Insulin (Sigma #I9278), 1 mM 3-isobutyl-1-methylxanthine (Sigma #I5879) in DMEM plus 10% FBS. Stimulation for adipogenic differentiation was started when cells reached full confluency: cells were grown for three days in induction medium, five days in maintenance medium (DMEM plus 10% FBS and 0.1 mg ml^−1 of insulin) and switched again to the induction medium for three additional days. After 14 days, adipocyte differentiation was assessed staining with by Oil Red-O (Sigma #O0625).

For angiogenesis assays in matrigel, starved MSCs or perivascular cells (PGFR+CD146+) isolated by FACs from mouse colons as indicated below, were cultured (1 × 10^4 per cm^2) for up to 20 h at 37 °C, 5% CO_2 on Growth Factor Reduced Matrigel Basement Membrane Matrix (BD Biosciences# 356231) using

coated 48 wells plates. For angiogenesis assays in 3D collagen gels, $2 \times 10^4$ starved MSCs were cultured for 2 days at 37 °C, 5% $CO_2$ on Collagen I Rat Tail (Corning #354236) using coated 48 wells plates and later cells were stained with Flourescein Phalloidin (Thermo Fisher#F432 ;1:40 1 h RT) to detect the lumens. Angiogenesis assays were quantified using an ImageJ-Fiji macro (http://image.bio.methods.free.fr/ImageJ/?Angiogenesis-Analyzer-for-ImageJ&lang=en).

**Cell migration and invasion assays.** Cell migration was assayed in 48-well modified Boyden chambers (BD Bioscience #353097) and cell invasion in matrigel invasion chambers (BD Bioscience 354480). The lower wells of the modified Boyden chamber were filled with DMEM 0.5% FBS or conditioned medium from WT or p38αΔ MSCs. Immortalized H5V endothelial cells[55] were tested for mycoplasm contamination and grown to 50–75% confluency, released with tryp-sin-EDTA, and resuspended in DMEM 0.5% FBS at a concentration of $10^6$ cells/ml. These cells were added to the upper wells at $2 \times 10^4$ cells/well and migration or invasion to the underside of the filters were measured after 6 h or 16 h at 37 °C, respectively.

**Co-cultures of MSCs and endothelial cells.** To evaluate the effect of MSCs on the ability of endothelial cells to form capillary-like structures, single-cell suspensions were seeded as mixtures. Co-cultures were established using a 1:1 ratio of H5V endothelial cells pre-stained with Cell tracker Red (Invitrogen #C34552) and MSCs pre-stained with Cell tracker Green (Invitrogen #C7025), and were grown in DMEM 10% FBS. Pictures were taken using a Confocal Microscope Leica SPE.

**AcLDL uptake assay.** MSCs were starved overnight and Alexa Fluor® 488 AcLDL (Invitrogen #L23380) was added and 4 h later cells were fixed and analyzed with a Confocal Microscope Leica SPE.

**Detection of TGF-β by ELISA.** Supernatants derived from MSCs were collected, centrifuged at $7500 \times g$ for 5 min at 4 °C to remove dead cells and cellular debris, and diluted 1:4 for analysis using the mouse Mouse TGF-β1 DuoSet ELISA kit following the manufacturer's instructions (R&D Systems # DY1679-05).

**Preparation of protein lysates and immunoblotting.** MSCs, colon perivascular cells were lysed in RIPA buffer containing 1% NP40, 150 mM NaCl, 50 mM Tris HCl pH 7.5, 2 mM EDTA, 2 mM EGTA, protease inhibitor cocktail (Roche, #11873580001), 20 mM sodium fluoride, 2 mM PMSF, 2 μM microcystin, 2 mM sodium orthovanadate, 1 mM DTT. Protein samples were quantified using the Bradford system (Sigma #B6916), and 40–80 μg of total protein lysates were fractionated by 10% or 12% SDS-PAGE and analyzed by immunoblotting. Alexa Fluro 680 or 800-conjugated secondary antibodies were used for 1 h at RT before visualization using the Odyssey Infrared Imaging System (Li-Cor, Biosciences). Anti-tubulin, GAPDH or Lamin A/C were used as loading controls. The primary and secondary antibodies and the dilutions used are listed in Supplementary Table 2.

**Immunohistochemistry and immunofluorescence.** For immunohistochemistry analysis, paraffin-embedded sections from mouse colons and AOM/DSS-induced tumors obtained as indicated above, and sections from colorectal tumor PDXs[20] were stained with the antibodies listed in Supplementary Table 3.

Immunofluorescence staining on paraffin sections was performed using the antibodies listed in Supplementary Table 3. The samples were visualized with Confocal Microscope Leica SPE or Zeiss LSM. Quantifications were performed using ImageJ-Fiji macro. See Code availability.

**Quantitative real-time PCR.** RNA was obtained from MSCs, colon fibroblasts or colon segments, using PureLink RNA kit (Ambion #12183018A) and treated with DNase (Roche #04716728001). RNA (1 μg) was reverse transcribed using a Super script II Reverse Transcriptase (Invitrogen #18064-014) with Random primers (Invitrogen #48190- 011). qRT-PCR was performed in triplicates using 50 ng of cDNA and SYBR green (Bio-Rad #1708886) on a Bio-Rad C1000 thermal cycler machine. Relative quantities (Δ cycle threshold values) were obtained by normal-izing against GAPDH. The primers are listed in Supplementary Table 4.

**Fractionation of cytoplasmic and nuclear proteins.** WT and p38αΔ MSCs were plated at subconfluent conditions on a 100 mm-tissue cultured dish. Cells were maintained with DMEM (0.5% FBS) and were either untreated or treated with TGF-β (5 ng ml$^{-1}$) for 20 h. The next day, cells were collected in 10 mM Hepes pH 7.5, 10 mM KCl, 5 mM MgCl$_2$ and protease inhibitors (Buffer A). Cell lysates were incubated for 15 min on ice. NP-40 was added to a final concentration of 0.3% followed by an incubation of 10 min on ice. Cell lysates were centrifuged at 3500 rpm and the supernatant was collected as cytoplasmic fraction. The pellet was washed once in Buffer A and was resuspended in 20 mM Hepes pH 7.5, 200 mM NaCl, 1% NP-40, protease inhibitors and DNAse 2-3 U per μl (Buffer B) and incubated for 30 min at 37 °C. Lysates were centrifuged at 14000 rpm and the supernatant was collected as nuclear fraction.

**TGF-β pathway reporter assay.** The synthetic CAGA-LUC reporter containing 12 tandem copies of the Smad-binding element[56] (a gift from Eduard Batlle, IRB Barcelona) was transfected in MSCs together with *Renilla reniformis* luciferase (RLluc). After 32 h, MSCs were treated with TGF-β (5 ng/ml) and 16 h later the promoter activity was measured. The reference control activity was measured using the Dual Luciferase Reporter Assay System (Promega #E1910), according to the manufacturer's instructions. Luc activity was normalized by RLluc activity.

**Analysis of *Mapk14* deletion.** Genomic DNA was isolated using standard phenol-chloroform extraction from sorted populations of PDGFRB$^+$ pericytes from colon tumors, and analyzed by qPCR with primers specific for exon 2 (floxed) and exon 12 (as a control) of the *Mapk14* gene encoding p38α. Relative amount of exon 2 versus exon 12 was determined. Primers are indicated in the two bottom rows of Supplementary Table 4.

**Colon disaggregation, mesenchymal cell isolation, and FACS analysis.** Colons were dissected, opened longitudinally and washed with cold PBS, incubated with 8 mM EDTA at 37 °C for 15 min. Supernatants were centrifuged at 1200 rpm for 5 min at 4 °C, and pelleted cells were digested with Dispase II (0.5 mg/ml, Roche #04942078001) at 37 °C for 25 min to isolated epithelial cells. To obtain lamina propria (mesenchymal and leukocytes cells) colon pieces after EDTA incubations were collected, cut into small pieces (2–3 mm) and digested with mix of collagenase A (1.75 mg/ml, Roche #10103586001) (0.05 mg/ml, Sigma #D4263) at 37 °C for 45 min. To quantify different cell populations, single-cell suspensions from both purifications were co-stained with pan-leukocyte antigen CD45-APC, CD31-PE-Cy7 for endothelial cells or PDGFRB-APC and CD146 PerCP/Cy5.5 for mesenchymal/perivascular cells, and analyzed on FACS Aria 2.0 (BD Biosciences). Antibodies are listed in the Supplementary Table 1. Cells were selected in the forward scatter/side scatter (FSC/SSC) dot plot and then gated to exclude cellular aggregates in the FSC/FSC dot plot. Gates for GFP and Tomato cells were set to compare the expression of endothelial and perivascular markers in different con-ditions. As negative control, a sample with no detectable fluorochrome expression was used.

**siRNA-mediated knockdown.** Validated siRNAs to target Mapk8 (JNK1) and Mapk9 (JNK2) were obtained from Ambion (#188651 and #73119). As a control, we used the siRNA Negative Control #1 (4390843). MSCs were transfected with 50 nM siRNAs using DharmaFECT-1 buffer (Cultek # T-2001-02) and 48 h later were analyzed by immunoblotting or in capillary-like tube formation assays.

**Microarray procedure.** Total RNA was extracted from MSCs using PureLink RNA kit (Ambion #12183018 A) and 25 ng were amplified using the TransPlex® Com-plete Whole Transcriptome Amplification Kit (WTA2, Sigma Aldrich). The cDNA (8 μg) was subsequently fragmented and labeled using GeneChip Human Mapping 250K Nsp Assay Kit (Affymetrix,Santa Clara, CA) according to manufacturer's instructions, and was hybridized to the Affymetrix® MG-430 PM Array for 16 h at 45 °C. Washing, staining and scanning of microarrays was performed using a GeneAtlas Fluidics station and scanner (Affymetrix).

**Microarray processing.** Microarray samples were processed using packages affy[57] and affyPLM from R[58] and Bioconductor[59]. Raw cel files were normalized using RMA background correction and summarization[60]. Standard quality controls were performed in order to identify abnormal samples and relevant sources of technical variability regarding: (a) spatial artifacts in the hybridization process (scan images and pseudo-images from probe level models); (b) intensity dependences of dif-ferences between chips (MvA plots); (c) RNA quality (RNA digest plot); (d) global intentisity levels (boxplot of perfect match log-intensity distributions before and after normalization and RLE plots); (e) anomalous intensity profile compared to the rest of samples (NUSE plots, Principal Component Analyses). No samples were excluded according to the results of these quality control checks. Chip probeset were annotated using the information provided by Affymetrix.

**Differential gene expression analysis.** Group comparisons in microarray experi-ments were performed using a linear model with empirical shrinkage[61] as imple-mented in the limma R package[62]. Potential sources of technical variability were included in these models as covariates (scanning batch for MSCs experiment; dataset for isolated cell populations from CRC primary tumors, see below). Adjustment by multiple contrasts was performed by the Benjamini-Hochberg method[63].

**Analysis of cell populations from human colorectal tumors.** Cell population profiles from human colorectal tumors were derived from two datasets publicly available in the Gene Expression Omnibus (GEO) repository:[64] GSE39395 and GSE39396. In these datasets, as described in ref. [10], FACS was used to separate the following cell populations from 14 fresh colorectal tumors: CD45$^+$/EpCAM$^-$/CD31$^-$/FAP$^-$; CD45$^-$/EpCAM$^+$/CD31$^-$/FAP$^-$; CD45$^-$/EpCAM$^-$/CD31$^+$/FAP$^-$; and CD45$^-$/EpCAM$^-$/CD31$^-$/FAP$^+$. Microarrays for each dataset were processed separately as described in Microarray processing section, and expression values were then corrected a-priori by biological sample. To homogenize the two expression

matrices, dataset GSE39395 was centered and scaled genewise to the mean and standard deviation of CD45+ and Epcam+ samples from GSE39396. Signatures of CAFs, FAP+ and endothelial cells (CD31+) were then derived from the resulting expression matrix. For doing so, we selected probesets overexpressed in the FAP+ and in the CD31+ samples each with a five minimum fold and FDR <5% compared to any other cell population (see Differential expression section). Genes mapping to selected probesets were translated to *Mus Musculus* homologous genes using the Mouse Genome Informatics (MGI) database[65] and entrez as gene identifier. To obtain a summary of FAP+ and CD31+ signatures in each sample of the MSCs dataset, expression values were centered and scaled gene-wise to produce z-scores, which were then averaged across all genes included in a given signature. The resulting score was in turn centered and scaled across samples. This procedure was carried out in the MSCs expression matrix after being corrected a-priori by scanning batch, using a linear model in which experimental group was included as covariate. Gene expression profiles were compared across experimental groups using a linear model in which scanning batch was included as covariate. A Wald test was used for statistical inference, and adjusted group means and corresponding 95% confidence intervals were derived from the model. Adjusted values, group means and standard errors derived from the models were represented graphically. A 5% level was set for statistical significance.

**Enrichment analysis**. Pathway enrichment was assessed through the preranked version of Geneset Enrichment Analysis (GSEA)[66]. GSEA was applied to the rankings defined by the t-statistic of the differential expression analysis described above (*Differential gene expression analysis* section). Genesets for analyses were derived from the KEGG pathway database[67] and from those annotated under Gene Ontology (GO)[68] terms as collected in the *org.Mm.eg.db* R package. GSEA was performed on each of these genesets collections separately. Expression data were summarized to gene level using probesets with the highest median absolute deviation within each gene (gene symbol).

**Statistical methods**. Data are presented as mean ± SEM. Statistical significance was determined by Student's *t* test using GraphPad Prism 7 software. *p* values < 0.05 were considered statistically significant. To assess group differences in qPCR data, a linear model was fitted to measured fold-changes in which experiment run was included as a covariate. Fold-changes were previously log2- transformed in order to fulfill the assumptions of the linear model. A Wald test was used to assess statistical significance, and adjusted group means and corresponding 95% confidence intervals were derived from the linear model. Within each analysis, *p*-values were adjusted by multiple comparisons using the Westfall method[69]. Values were graphically represented in log2 scale after a-priori correction for experiment run using the corresponding linear model. Adjusted means and standard errors derived from the models were also displayed in the graphics. A 5% level was set for statistical significance.

**Reporting summary**. Further information on research design is available in the Nature Research Reporting Summary linked to this article.

## Data availability
Microarray data have been deposited in the Gene Expression Omnibus (GEO) under the accession code GSE83810. The source data underlying Figs. 1–8 and Supplementary Figs. 1–8 as well as the raw images for the immunoblots are provided in the Source Data file. All other data supporting the findings of this study are available from the corresponding author on reasonable request.

## Code availability
The macros used for image analysis and the programming codes are available on reasonable request.

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

## Acknowledgements

We are grateful to Gustavo Leone (Ohio State University, Columbus, USA) for providing FSP1-Cre mice, Ralf Adams (Max Planck Institute for Molecular Biomedicine, Muenster, Germany) for PDGFRB-Cre-ERT2 mice, Eric Brown (University of Pennsylvania School of Medicine, Philadelphia, USA) for UBC-Cre-ERT2 mice, and Manuel Hidalgo (CNIO, Madrid) for PDX samples. We thank the Genomics and Advanced Digital Microscopy Core Facilities of IRB Barcelona for excellent technical assistance, Sebastien Tosi and Anna Lladó for generating the macros used for image quantifications, and Evarist Planet for initial analysis of gene expression data. We are grateful to Eduard Batlle (IRB Barcelona), Ralf Adams (Max Planck Institute Muenster) and Antonio Garcia de Herreros (IMIM Barcelona) for critically reading the manuscript. This work was supported by grants from the European Commission (Advanced ERC 294665), Fundación Olga Torres, Maraató-TV3 (20133430), MINECO (SAF2016-81043-R) and AGAUR (2017 SGR-557). IRB Barcelona is the recipient of institutional funding from MINECO (Government of Spain) through the Centres of Excellence Severo Ochoa award and from the CERCA Program of the Catalan Government.

## Author contributions

R.B. designed and performed most of the experiments, analyzed data, prepared the figures and wrote the manuscript. E.A. provided technical assistance with in vivo and in vitro experiments. L.G., E.L., A.I., and N.G. performed some biochemical experiments, mouse treatments, immunohistochemistry and quantifications. A.B.-L. performed bioinformatics and statistical analyses. A.R.N. provided funding, supervised the overall study, designed experiments, analyzed data, and wrote the manuscript.

## Additional information

**Competing interests:** The authors declare no competing interests.

