## [Peer Review File · Nature Communications]

Editorial Note: This manuscript has been previously reviewed at another journal that is not operating a transparent peer review scheme. This document only contains reviewer comments and rebuttal letters for versions considered at Nature Communications. Mentions of the other journal have been redacted.

Reviewers' Comments:

Reviewer #1:

Remarks to the Author:

The authors have addressed my concerns.

Reviewer #2:

Remarks to the Author:

In this work, Battle et al., describe the p38a-mediated transition of mesenchymal stem cells to endothelial cells. The revised manuscript is significantly improved and the authors have addressed either with additional experiments, or with changes in the text, all my previous comments.

Reviewer #3:

Remarks to the Author:

The authors answered to my criticisms. There might be still some aspects unclear but the paper is worth to be published in Nature Comm.

Manuscript NCOMMS-18-38330A. Response to Reviewers.**Reviewer #1**

The authors have addressed my concerns.

Reviewer #2

In this work, Battle et al., describe the p38a-mediated transition of mesenchymal stem cells to endothelial cells. The revised manuscript is significantly improved and the authors have addressed either with additional experiments, or with changes in the text, all my previous comments.

Reviewer #3

The authors answered to my criticisms. There might be still some aspects unclear but the paper is worth to be published in Nature Comm.

The three reviewers have no additional requests and recommended publication of the manuscript.